# A Rigorous Study Of The Deep Taylor Decomposition

**Leon Sixt**                                                                     *leon.sixt@fu-berlin.de*
*Department of Computer Science*
*Freie Universität Berlin*

**Tim Landgraf**                                                                  *tim.landgraf@fu-berlin.de*
*Department of Computer Science*
*Freie Universität Berlin*

**Reviewed on OpenReview:** *https: // openreview. net/ forum? id= Y4mgmw9OgV*

**Code Repository:** *https: // github. com/ berleon/ A-Rigorous-Study-Of-The-Deep-Taylor-Decomposition*

## Abstract

Saliency methods attempt to explain deep neural networks by highlighting the most salient features of a sample. Some widely used methods are based on a theoretical framework called Deep Taylor Decomposition (DTD), which formalizes the recursive application of the Taylor Theorem to the network's layers. However, recent work has found these methods to be independent of the network's deeper layers and appear to respond only to lower-level image structure. Here, we investigate the DTD theory to better understand this perplexing behavior and found that the Deep Taylor Decomposition is equivalent to the basic gradient×input method when the Taylor root points (an important parameter of the algorithm chosen by the user) are locally constant. If the root points are locally input-dependent, then one can justify any explanation. In this case, the theory is under-constrained. In an empirical evaluation, we find that DTD roots do not lie in the same linear regions as the input – contrary to a fundamental assumption of the Taylor theorem. The theoretical foundations of DTD were cited as a source of reliability for the explanations. However, our findings urge caution in making such claims.

## 1 Introduction

Post-hoc explanations are popular for explaining Machine Learning models as they do not require changing the model's architecture or training procedure. In particular, feature attribution methods are widely used. They assign a saliency score to each input dimension, reflecting their relevance for the model's output. For images, the saliency scores can be visualized as heatmaps (see Figure 2).

Evaluating post-hoc explanations is challenging because it is inherently circular: As we do not understand the internal workings of the model, which we are trying to explain, we cannot judge the quality of the explanation. The situation is further complicated as many methods simplify the model's complexity to render explanations accessible to the human eye. For example, most methods focus on the local neighborhood of an input sample, and rely on assumptions such as linearity (e.g. gradient-based methods) or independence of the input features (e.g. approximation of Shapley values (Štrumbelj & Kononenko, 2011; 2014; Lundberg & Lee, 2017; Kumar et al., 2020)).

These factors and the complexity of deep neural networks make it difficult to assess whether an explanation is correct or not. We can not disentangle failures of the explanation method and unexpected behavior of the model. While it is acceptable for methods to introduce simplifications or rely on assumptions, their existence, purpose, and violation should be made transparent. In the best case, a method would be based on a solid theoretical foundation providing guarantees regarding an explanation's correctness.

Such a theoretical foundation is the *Deep Taylor Decomposition* (DTD, Montavon et al. (2017)). DTD recursively applies the Taylor Theorem to the network's layers, and backpropagates modified gradients to the input, thereby computing the input's relevance. It was used as theoretical foundation of LRP (Bach et al., 2015), a poplar method to explain image models. LRP was repeatedly advertised as a sound and reliable explanation technique (Montavon et al., 2019; Samek et al., 2021b; Holzinger et al., 2022). For example, Holzinger et al. (2022) stated: *"The main advantages of LRP are its high computational efficiency [...], its theoretical underpinning making it a trustworthy and robust explanation method [...], and its long tradition and high popularity [...]."*

However, Sixt et al. (2020) has shown that certain LRP and DTD backpropagation rules create explanations partially independent of the model's parameters: the explanation will remain the same even if the last layer's parameters are randomized. The theoretical analysis in (Sixt et al., 2020) revealed that the propagation matrices, which correspond to the layers' Jacobian matrices, are all positive and their product converges to a rank-1 matrix quickly. To obtain the saliency map, the result is usually normalized, and thereby even the last single degree of freedom is lost. Thus, the explanation does not change when explaining a different class or the parameters of the deeper layers is changed.

This perplexing behavior questions the consistency of DTD directly. While Sixt et al. (2020) described the convergence to the rank-1 matrix in detail, the failure was not related to DTD's theory such as the choice of root points[1] and the recursive application of the Taylor Theorem. Here, we fill this gap: *Can we identify flaws in DTD's theory that would explain the perplexing behavior of ignoring the network's parameters? Does DTD provide transparency regarding its assumptions and guarantees about the explanation's correctness?*

Before we approach these questions, we summarize the relevant background of the Deep Taylor Decomposition in Section 3. For completeness, we start with the well-known Taylor theorem and then discuss how the theorem connects to DTD's relevances. We then continue with stating the recursive application of the Taylor Theorem formally and recapitulate the so-called *train-free DTD* approximation, which allows to compute layer-wise relevances efficiently.

In section 4, we present our theoretical analysis of DTD. In particular, we contribute: **(C1)** a proof that the root points must be contained in the same linear region as the input; **(C2)** we generalize a previous observation about $LRP_0$ (Shrikumar et al., 2016): if the layers' root points are chosen locally constant w.r.t the layers' input, then DTD's relevances take a similar form as input×gradient; **(C3)** DTD is under-constrained: if the root points depend on the layers' input, then the Deep Taylor Decomposition can be used to create any arbitrary explanation; **(C4)** we also find that DTD cannot be extended easily to analytic activation functions (e.g. Softplus), without introducing complex higher-order derivatives with the same order as the number of network layers.

In an empirical evaluation (Section 5), we applied the theoretical insights from the previous section and studied the *train-free DTD* approximation in several experiments: **(C5)** The train-free DTD does not enforce the root points to be located in the valid local linear region of the network; **(C6)** We also validated this empirically using a small multilayered perceptron, where we found a substantial number of samples having roots located outside the valid local linear region; **(C7)** Additionally, we include a reproducibility study of (Arras et al., 2022) that claimed that DTD's explanations would not suffer from the problems reported in Sixt et al. (2020). This reproducibility study also highlights DTD's black-box character and how difficult it is to evaluate explanation quality empirically.

Given the theoretical and empirical evidence, we conclude that DTD obscures its simplifications and violates its own assumptions. DTD is underconstrained and even allows justifying virtually any explanation.

## 2   Related Work

The theoretical analysis of explanation methods is a small research area. PatternAttribution (Kindermans et al., 2018) investigated the insensitiveness of DTD rules to input noise and then proposed a way to learn

---

[1] Following Montavon et al. (2017), we name the points used for the Taylor Theorem *root points*. For example, for a function $f : \mathbb{R} \to \mathbb{R}$, the first-order Taylor approximation is $f(x) \approx f(\tilde{x}) + f'(\tilde{x})(x - \tilde{x})$, where $\tilde{x}$ is the root point.

the root points from data. Other lines of work are the manipulation of saliency maps (Dombrowski et al., 2019; Viering et al., 2019; Wang et al., 2020), the runtime-complexity of explanation methods (Waeldchen et al., 2021), or the explanations methods with provable guarantees (Chen et al., 2019).

Previous works have analyzed the theoretical properties of various saliency methods. For example, the insensitivity of Guided-Backprop (Springenberg et al., 2014) was analyzed in (Nie et al., 2018), and (Lundstrom et al., 2022) found flaws in the theoretical motivation of integrated gradients Sundararajan et al. (2017). (Kumar et al., 2020) discussed the issues from an independence assumption between input variables, often introduced in sampling algorithms (Štrumbelj & Kononenko, 2011; 2014; Lundberg & Lee, 2017). In (Shah et al., 2021), it was empirically analyzed and proven for a specific dataset that the Gradient's magnitude will not correspond to relevant features. Our work also analyzes the theoretical properties of saliency methods but differs from previous works as it focuses on the DTD and LRP methods.

Although different review articles (Montavon et al., 2018; 2019; Samek et al., 2021b;a) and extensions of LRP and Deep Taylor (Binder et al., 2016; Kohlbrenner et al., 2020; Hui & Binder, 2019; Ali et al., 2022) have been published, none discussed the theoretical issues brought forward in our manuscript.

## 3 Background

In this section, we provide the necessary background on Deep Taylor Decomposition to understand the theoretical analysis in Section 4. We mainly reproduce the derivations given in (Montavon et al., 2017; 2018). If we comment on the derivations, we do this in the *Remark* sections.

### 3.1 Taylor Theorem for multivariate functions

Taylor Theorem for multivariate functions can be concisely stated using multi-index notation. A multi-index $\boldsymbol{\alpha} \in \mathbb{N}_0^k$ is a vector of non-negative integers ($\boldsymbol{\alpha} = [\alpha_1, \dots, \alpha_k]$). The following operations are defined as: $\boldsymbol{\alpha}! = \alpha_1!\alpha_2!\dots\alpha_k!$, $|\boldsymbol{\alpha}| = \sum_{i=1}^{k}\alpha_i$, $\boldsymbol{x}^{\boldsymbol{\alpha}} = x_1^{\alpha_1}x_2^{\alpha_2}\dots x_k^{\alpha_k}$, and $\partial^{\boldsymbol{\alpha}}f = \partial^{|\boldsymbol{\alpha}|}f/(\partial^{\alpha_1}x_1\partial^{\alpha_2}x_2\dots\partial^{\alpha_k}x_k)$, where $\boldsymbol{x} \in R^k$ and $f : \mathbb{R}^k \to \mathbb{R}$. The following theorem is adapted from Folland (2002, Theorem 2.68):

**Theorem 1** (Multivariate Taylor Theorem). *Suppose $f : \mathbb{R}^d \to \mathbb{R}$ is of differentiability class $C^k$ on an open convex set $S$. If $\boldsymbol{x} \in S$ and $\tilde{\boldsymbol{x}} \in S$, then:*

$$f(\boldsymbol{x}) = \sum_{|\boldsymbol{\alpha}| \leqslant k} \frac{\partial^{\boldsymbol{\alpha}} f(\tilde{\boldsymbol{x}})}{\boldsymbol{\alpha}!}(\boldsymbol{x} - \tilde{\boldsymbol{x}})^{\boldsymbol{\alpha}} + g_k(\boldsymbol{x}, \tilde{\boldsymbol{x}}), \tag{1}$$

*where the remainder is given by:*

$$g_k(\boldsymbol{x}, \tilde{\boldsymbol{x}}) = k \sum_{|\boldsymbol{\alpha}|=k} \frac{(\boldsymbol{x} - \tilde{\boldsymbol{x}})^{\boldsymbol{\alpha}}}{\boldsymbol{\alpha}!} \int_0^1 (1-t)^{k-1} \Big[ \partial^{\boldsymbol{\alpha}} f\big(t\boldsymbol{x} + (1-t)\,\tilde{\boldsymbol{x}}\big) - \partial^{\boldsymbol{\alpha}} f(\boldsymbol{x}) \Big] dt. \tag{2}$$

As the Deep Taylor Decomposition focuses on neural networks with ReLU activations, we will mainly look at the first-order Taylor Theorem:

$$f(\boldsymbol{x}) = f(\tilde{\boldsymbol{x}}) + \frac{\partial f(\boldsymbol{x})}{\partial \boldsymbol{x}}\Big|_{\boldsymbol{x}=\tilde{\boldsymbol{x}}} \cdot (\boldsymbol{x} - \tilde{\boldsymbol{x}}) \tag{3}$$

where $\tilde{\boldsymbol{x}}$ is the root point, and $|_{\boldsymbol{x}=\tilde{\boldsymbol{x}}}$ denotes the gradient evaluated at the root point $\tilde{\boldsymbol{x}}$. The higher order terms are zero due to the local linearity of ReLU networks. As the Taylor Theorem requires $f \in C^1$ (i.e., all partial derivatives $\partial f(\boldsymbol{x})/\partial \boldsymbol{x}$ must be continuous in the local neighborhood $S$), the root point must be within the same linear region as the input.

**Definition 1** (Linear Region). A linear region of a function $f : \mathbb{R}^d \to \mathbb{R}$ is the set $N_f(\boldsymbol{x})$ of all points $\boldsymbol{x}' \in N_f(\boldsymbol{x})$ that (1) have the same gradient at $\boldsymbol{x}$: $\nabla f(\boldsymbol{x}) = \nabla f(\boldsymbol{x}')$, and (2) can be reached from $\boldsymbol{x}$ without passing through a point $\boldsymbol{a}$ with a different gradient, i.e., $\nabla f(\boldsymbol{a}) \neq \nabla f(\boldsymbol{x})$.

In case of a ReLU network, the approximation error cannot be bounded when selecting a root point $\tilde{\boldsymbol{x}} \notin N_f(\boldsymbol{x})$, as that linear region's gradient might differ substantially.

### 3.2 Taylor Theorem and Relevances

In the previous section, we have recapitulated the Taylor Theorem. We now discuss how the Taylor Theorem can be used to compute input relevances. A common approach explaining deep neural network is to contrast the network's output with a similar point predicted differently. A user could then study pairs $(\boldsymbol{x}, f(\boldsymbol{x}))$ and $(\tilde{\boldsymbol{x}}, f(\tilde{\boldsymbol{x}}))$ and relate the input differences to the output differences. To guide the user's attention, it would be desirable to highlight which changes between $\boldsymbol{x}$ and $\tilde{\boldsymbol{x}}$ were responsible for the difference in the output. The first observation in (Montavon et al., 2017) is that the network's output differences can be redistributed to the input by using the Taylor Theorem. If the point $\tilde{\boldsymbol{x}}$ is in the local neighborhood $N_f(\boldsymbol{x})$, we can use the first-order Taylor Theorem (equation 3) to write the difference $f(\boldsymbol{x}) - f(\tilde{\boldsymbol{x}})$ as:

$$f(\boldsymbol{x}) - f(\tilde{\boldsymbol{x}}) = \frac{\partial f(\boldsymbol{x})}{\partial \boldsymbol{x}}\Big|_{\boldsymbol{x}=\tilde{\boldsymbol{x}}} \cdot (\boldsymbol{x} - \tilde{\boldsymbol{x}}) \tag{4}$$

The relevance of the input $R : \mathbb{R}^d \to \mathbb{R}^d$ is then defined to be the point-wise product of the partial derivatives with the input differences:

$$R(\boldsymbol{x}) = \frac{\partial f(\boldsymbol{x})}{\partial \boldsymbol{x}}\Big|_{\boldsymbol{x}=\tilde{\boldsymbol{x}}} \odot (\boldsymbol{x} - \tilde{\boldsymbol{x}}) \tag{5}$$

While this would be a simple way to compute the relevances, the following reasons are given in (Montavon et al., 2017; 2019), to not directly use the Taylor Theorem on the network output:

1. **Adversarial perturbations** (Szegedy et al., 2013): Small input perturbations can lead to a large change in the output. Therefore, the difference in the output might be enormous but $|\boldsymbol{x} - \tilde{\boldsymbol{x}}|$ tiny and uninterpretable.

2. **Finding a root point might be difficult**: *"It is also not necessarily solvable due to the possible non-convexity of the minimization problem"* (Montavon et al., 2017).

3. **Shattered gradients** (Balduzzi et al. (2017)): *"While the function value $f(x)$ is generally accurate, the gradient of the function is noisy"* (Montavon et al., 2019).

**Remark 1.** We want to point out that the more general problem seems to be that the local linear regions are tiny, or rather the number of linear regions grows exponentially with the depth of the network in the worst case (Arora et al., 2018; Xiong et al., 2020; Montufar et al., 2014). This restricts the valid region for the root point to a small neighborhood around the input.

### 3.3 Deep Taylor: Recursive Application of Taylor Theorem

The main idea of (Montavon et al., 2017) is to recursively apply the Taylor Theorem to each network layer. Before we present this in detail, we first we need to clarify the notation of an n-layered ReLU network shortly:

**Definition 2** (ReLU network). An $n$-layered ReLU network $f : \mathbb{R}^{d_1} \to \mathbb{R}^{d_{n+1}}_{\geqslant 0}$ is the composition of $n$ functions $f = f_n \circ \ldots \circ f_1$, where each function $f_l : \mathbb{R}^{d_l} \to \mathbb{R}^{d_{l+1}}_{\geqslant 0}$ has the form $f_l(\boldsymbol{a}_l) = [W_l \boldsymbol{a}_l]^+$, and where $[.]^+$ is the ReLU activation.

Instead of directly calculating the relevance of the input as done in the previous section, we can apply Taylor Theorem to the final network layer and then apply the Taylor Theorem again to the resulting relevance. By recursively applying the Taylor Theorem per individual layer, we can calculate the relevance of the input. As the base case of the recursive application, the relevance of the network output is set to the value of the explained logit $f_{[\xi]}(\boldsymbol{x}) = \boldsymbol{a}_{n+1_{[\xi]}}$:

$$R^{n+1}(\boldsymbol{a}_{n+1}) = \boldsymbol{a}_{n+1_{[\xi]}}, \tag{6}$$

where $R^{n+1}$ denotes the relevance of the $n + 1$-th network activation. We decided to use superscripts for the relevance functions as their individual dimensions are often index as in $R^{n+1}_{[j]}$. Suppose that we already

know the relevance function $R^{l+1}(\boldsymbol{a}_{l+1}) \in \mathbb{R}^{d_{l+1}}$ for the layer $l+1$. We can then calculate the relevance of $\boldsymbol{a}_l$ (the input to layer $l$) to the $j$-th coordinate of $R^{l+1}(\boldsymbol{a}_{l+1})$:

$$R^{l+1}_{[j]}(\boldsymbol{a}_{l+1}) = R^{l+1}_{[j]}(f_l(\tilde{\boldsymbol{a}}_l)) + \left.\frac{\partial R^{l+1}_{[j]}(f_l(\boldsymbol{a}_l))}{\partial \boldsymbol{a}_l}\right|_{\boldsymbol{a}_l=\tilde{\boldsymbol{a}}_l(\boldsymbol{a}_l)} \cdot (\boldsymbol{a}_l - \tilde{\boldsymbol{a}}_l(\boldsymbol{a}_l)), \tag{7}$$

where we used $\boldsymbol{a}_{l+1} = f_l(\boldsymbol{a}_l)$. The root point is selected in dependency of the layers' input $\boldsymbol{a}_l$, i.e., it is a function $\tilde{\boldsymbol{a}}_l : \mathbb{R}^{d_l} \to \mathbb{R}^{d_l}$. The total relevance of the input to layer $l$ is given by the sum over all $d_{l+1}$ hidden neurons.

**Definition 3** (Recursive Taylor). Given a function $f : \mathbb{R}^{d_1} \to \mathbb{R}^{d_{n+1}}$, which can be written as a composition of $n$ functions $f = f_1 \circ \ldots \circ f_n$ with $f_l : \mathbb{R}^{d_l} \to \mathbb{R}^{d_{l+1}}$, the input to each function $f_l$ are denoted by $\boldsymbol{a}_l$ and $\boldsymbol{a}_{n+1}$ specifcs $f$'s output. Additionally, a root point function $\tilde{\boldsymbol{a}}_l : \mathbb{R}^{d_l} \to \mathbb{R}^{d_l}$ is defined for each layer, which must only return admissible values $\tilde{\boldsymbol{a}}_l(\boldsymbol{a}_l) \in N_{R_l}(\boldsymbol{x})$ and $\tilde{\boldsymbol{a}}_l(\boldsymbol{a}_l) \neq \boldsymbol{a}_l$. Then, the base case is given by $R^{n+1}(\boldsymbol{a}_{n+1}) = \boldsymbol{a}_{n+1_{[\xi]}}$ and the relevance function $R^l : \mathbb{R}^{d_l} \to \mathbb{R}^{d_l}$ of layer $l \neq n$ is recursively defined by:

$$R^l(\boldsymbol{a}_l) = \sum_{j=1}^{d_{l+1}} \left( \left.\frac{\partial R^{l+1}_{[j]}(f_l(\boldsymbol{a}_l))}{\partial \boldsymbol{a}_l}\right|_{\boldsymbol{a}_l=\tilde{\boldsymbol{a}}_l^{(j)}(\boldsymbol{a}_l)} \odot \left( \boldsymbol{a}_l - \tilde{\boldsymbol{a}}_l^{(j)}(\boldsymbol{a}_l) \right) \right) \tag{8}$$

The above definition corresponds to equation 6 in (Montavon et al., 2017). Except for the root point selection, which will be discussed in the following sections, definition 3 contains all information to implement the recursive decomposition using an automatic differentiation library. An exemplary pseudo-code can be found in Algorithm 1. Before continuing with the approximations of the Deep Taylor Decomposition, we want to make a few remarks:

**Remark 2** (No Axiomatic Motivation). Only some vague arguments are provided to motivate the recursive application of the Taylor Theorem:

> *The deep Taylor decomposition method is inspired by the divide-and-conquer paradigm, and exploits the property that the function learned by a deep network is decomposed into a set of simpler subfunctions, either enforced structurally by the neural network connectivity, or occurring as a result of training.* – Montavon et al. (2017, Section 3)

In contrast, Shapely values (Shapley, 1951) are motivated by four axiomatic properties, which are uniquely fulfilled by the Shapely values. A comparable set of axioms with uniqueness result does not exist for the Deep Taylor Decomposition.

### 3.4 Deep Taylor Decomposition For A One-Layer Network and DTD's rules

In the previous section, we introduced the recursive application of the Taylor Theorem. For a concrete example, we will now discuss how DTD is applied to a one-layered network. It will also explain the propagation rules of DTD. This subsection corresponds to Section 4 in Montavon et al. (2017), and we will refer to the corresponding equations with the notation (DTD eq. 11).

The one-layered network consists of a linear layer with a ReLU activation followed by a sum-pooling layer, i.e. $f : \mathbb{R}^d \to \mathbb{R}$, $f(\boldsymbol{x}) = \sum_j [W\boldsymbol{x} + \boldsymbol{b}]_j^+$, where $[.]^+$ is the ReLU activation. We will denote the output of the ReLU layer as $\boldsymbol{h}(\boldsymbol{x}) = [W\boldsymbol{x} + \boldsymbol{b}]^+$, and the sum-pooling layer as $y(\boldsymbol{h}) = \sum_j \boldsymbol{h}_j$. For this subsection, we will denote the relevance function with $R^{\boldsymbol{x}}$, $R^{\boldsymbol{h}}$, and $R^y$ for the input, hidden, and output layer, respectively.

The relevance of the final layer is simply given by the network's output (equation 6; DTD eq. 8):

$$R^y(\boldsymbol{h}(\boldsymbol{x})) = y(\boldsymbol{h}(\boldsymbol{x})). \tag{9}$$

DTD suggests to select a root point $\tilde{\boldsymbol{h}}$ such that $y(\tilde{\boldsymbol{h}}) = 0$. The advantage of $y(\tilde{\boldsymbol{h}}) = 0$ is that the network's output $y(\boldsymbol{h})$ is absorbed to the first-order term (i.e., $f(\tilde{\boldsymbol{x}}) = 0$ in equation 3). We then have $y(\boldsymbol{h}) = \frac{\partial R^y(\tilde{\boldsymbol{h}})}{\partial \tilde{\boldsymbol{h}}} \cdot (\boldsymbol{h} - \tilde{\boldsymbol{h}})$ such that the network output is fully redistributed to the hidden layer's relevance. Additionally,

the root point should be a valid input to the layer. As $y(\boldsymbol{h})$'s input comes from the ReLU layer $\boldsymbol{h}(\boldsymbol{x})$, it is positive and only $\tilde{\boldsymbol{h}} = 0$ solves $\sum_j \tilde{\boldsymbol{h}}_j = 0$. The derivative $\partial R_{\boldsymbol{h}}^f(\tilde{\boldsymbol{h}})/\partial \tilde{\boldsymbol{h}} = \partial y(\tilde{\boldsymbol{h}})/\partial \tilde{\boldsymbol{h}} = 1$ and therefore, we can use equation 8 to write the relevance of the ReLU layer's output as (DTD eq. 10):

$$R^{\boldsymbol{h}}(\boldsymbol{x}) = \left.\frac{\partial R^y(\boldsymbol{h})}{\partial \boldsymbol{h}}\right|_{\tilde{\boldsymbol{h}}=0} \odot \boldsymbol{h} = \boldsymbol{h} \tag{10}$$

As $\boldsymbol{h}$ is the ReLU output, we can write $R^{\boldsymbol{h}}(\boldsymbol{x})$ also as (DTD eq. 11):

$$R^{\boldsymbol{h}}(\boldsymbol{x}) = [W\boldsymbol{x} + \boldsymbol{b}]^+ \tag{11}$$

The next step is to connect the input relevance $R^{\boldsymbol{x}}(\boldsymbol{x})$ with the ReLU neurons' relevances $R^{\boldsymbol{h}}$. We will use equation 8 and apply the Taylor theorem to the relevance of each hidden neuron $\boldsymbol{h}_{[j]}$:

$$R^{\boldsymbol{x}}(\boldsymbol{x}) = \sum_{j=1}^d \left( \left.\frac{\partial R_{[j]}^{\boldsymbol{h}}(\boldsymbol{x})}{\partial \boldsymbol{x}}\right|_{\boldsymbol{x}=\tilde{\boldsymbol{x}}^{(j)}} \odot (\boldsymbol{x} - \tilde{\boldsymbol{x}}^{(j)}) \right) = \sum_{j=1}^d \left( \boldsymbol{w}_j \odot (\boldsymbol{x} - \tilde{\boldsymbol{x}}^{(j)}) \right), \tag{12}$$

where we used that the derivative of the hidden neuron $\boldsymbol{h}_{[j]}$ w.r.t the input is the weight vector $\boldsymbol{w}_j = W_{[j:]}$.

**Relevance Propagation Rules** For the root point $\tilde{\boldsymbol{x}}$, we could, in theory, select any point in the half-space $\boldsymbol{w}_j\tilde{\boldsymbol{x}} + b_j > 0$, as they are all valid according to the Taylor Theorem. However, as it is beneficial to fully redistribute the relevance, DTD proposed selecting a point that sets the $j$-th neuron relevance to zero, i.e., any point on the hyperplane $\boldsymbol{w}_j^T \tilde{\boldsymbol{x}} + \boldsymbol{b}_j = 0$. The non-differentiability at the ReLU hinge is resolved by picking the gradient from the case $\boldsymbol{w}_j\tilde{\boldsymbol{x}} + b_j > 0$.

As there is no unique solution, DTD derives different points by starting at the input $\boldsymbol{x}$ and moving along a direction $\boldsymbol{v}_j$ such that $\tilde{\boldsymbol{h}}_j = \boldsymbol{x} - t\boldsymbol{v}_j$ with $t \in \mathbb{R}$. The root point $\tilde{\boldsymbol{x}}_j$ is then the intersection of the line $\boldsymbol{x} + t\boldsymbol{v}_j$ with the hyperplane $\boldsymbol{w}_j^T \boldsymbol{x} + \boldsymbol{b}_j = 0$. Combining these equations yields $t = -\frac{\boldsymbol{w}_j^T \boldsymbol{x} + \boldsymbol{b}_j}{\boldsymbol{w}_j^T \boldsymbol{v}_j}$ and therefore the root point $\tilde{\boldsymbol{x}}^{(j)} = \boldsymbol{x} - \frac{\boldsymbol{w}_j^T \boldsymbol{x} + \boldsymbol{b}_j}{\boldsymbol{w}_j^T \boldsymbol{v}_j}\boldsymbol{v}_j$. Substituting this into equation 12 yields: (DTD-Appendix eq. 6, 7)

$$R^{\boldsymbol{x}}(\boldsymbol{x}) = \sum_{j=1}^d \left( \boldsymbol{w}_j \odot \frac{\boldsymbol{w}_j^T \boldsymbol{x} + \boldsymbol{b}_j}{\boldsymbol{w}_j^T \boldsymbol{v}_j}\boldsymbol{v}_j \right) = \sum_{j=1}^d \left( \boldsymbol{w}_j \odot \frac{\boldsymbol{v}_j}{\boldsymbol{w}_j^T \boldsymbol{v}_j}R_{[j]}^{\boldsymbol{h}}(\boldsymbol{x}) \right). \tag{13}$$

While almost all choices of $\boldsymbol{v}$ yield a root point with $R^{\boldsymbol{x}}(\tilde{\boldsymbol{x}}) = 0$ (except $\boldsymbol{v} \perp \boldsymbol{w}$), a few special directions exists:

- *The $w2$-rule* chooses the closest root point in L2 metric: $\boldsymbol{v}_j = \boldsymbol{w}_j$. This will yield the root point $\tilde{\boldsymbol{x}} = \boldsymbol{x} - \frac{\boldsymbol{w}_j}{\boldsymbol{w}_j^T \boldsymbol{w}_j}R^{\boldsymbol{h}}(x)$ and the following relevance propagation rule: $R^x(\boldsymbol{x}) = \sum_{j=1}^d \frac{\boldsymbol{w}_j^2}{\boldsymbol{w}_j^T \boldsymbol{w}_j}R^{\boldsymbol{h}}(\boldsymbol{x})$, where $\boldsymbol{w}_j^2 = \boldsymbol{w}_j \odot \boldsymbol{w}_j$.

- *The $z^+$ rule* uses a direction that always yields a positive root: $\boldsymbol{v}_j = \mathbb{1}_{\boldsymbol{w}_j \geqslant 0}\boldsymbol{x}$, which is preferred for positive inputs (e.g. from ReLU activations). The resulting root point is $\tilde{\boldsymbol{x}} = \boldsymbol{x} - \frac{\boldsymbol{w}\mathbb{1}_{\boldsymbol{w}_j \geqslant 0}\boldsymbol{x}}{\boldsymbol{w}_j^T(\mathbb{1}_{\boldsymbol{w}_j \geqslant 0}\boldsymbol{x})}R_{[j]}^{\boldsymbol{h}}(x)$ and the following relevance propagation rule: $R^x(\boldsymbol{x}) = \sum_{j=1}^d \frac{\boldsymbol{z}_j^+}{\sum_i \boldsymbol{z}_{ji}^+}R_{[j]}^{\boldsymbol{h}}(\boldsymbol{x})$, where $\boldsymbol{z}_j^+ = \mathbb{1}_{\boldsymbol{w}_j \geqslant 0}\boldsymbol{x} \odot \boldsymbol{w}_j^+$.

- *The gamma rule* proposed in Montavon et al. (2019) uses the search direction $\boldsymbol{v}_j = 1 + \gamma\mathbb{1}_{\boldsymbol{w}_j \geqslant 0} \odot \boldsymbol{x}$, where $\gamma \in \mathbb{R}^+$. The corresponding relevance propagation rule is then: $R^{\boldsymbol{x}}(\boldsymbol{x}) = \sum_{j=1}^d \frac{\boldsymbol{w}_j + \gamma\boldsymbol{z}_j^+}{\boldsymbol{w}_j^T(1 + \gamma\boldsymbol{z}_j^+)}R_{[j]}^{\boldsymbol{h}}(\boldsymbol{x})$. In the limit $\gamma \to \infty$, the gamma rule becomes the $z^+$-rule.

- A special case is the $LRP_0$ rule which does not use any vector to find a root point but chooses $\tilde{\boldsymbol{x}} = 0$. Although zero is not a valid root point in general, it was shown that $LRP_0$ corresponds to gradient×input Shrikumar et al. (2016); Ancona et al. (2018); Kindermans et al. (2016). The $LRP_\varepsilon$ rule is an extension of $LRP_0$ that adds a small $\varepsilon$ to increase numeric stability.

### 3.4.1 Which rule should be chosen?

The computed relevance values depend substantially on the rule. For example, the $\mathrm{LRP}_0$ rule can compute negative relevance values, whereas the $z^+$ rule will always return positive relevance values. In Montavon et al. (2017, Sec. 4.1, 4.2, 4.3), the input domain was used as the primary selection criterion. For example, it was suggested to pick $z^+$ rule for $\mathbb{R}_0^+$ and the $w^2$ rule for the domain $\mathbb{R}$. The input domain is not a sufficient selection for the root points, as it does not provide a unique solution. In the later work Montavon et al. (2019), other selection criterion were proposed for deep neural networks, which we will analyze in Section 3.5.1. For now, we can conclude that no principled way to pick the roots and rules exists.

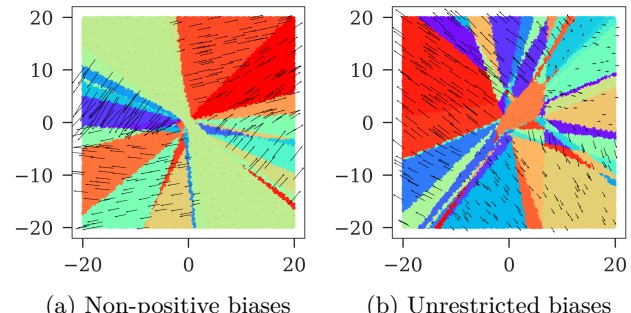

(a) Non-positive biases          (b) Unrestricted biases

Figure 1: Local linear regions of an randomly initialized neural network (3 layers, ReLU, 2 inputs, 10 hidden neurons). The biases are initialized **(a)** non-positive and **(b)** unrestricted. The gradient are visualized as arrows for a random selection of points.

### 3.4.2 Non-positive biases

In Montavon et al. (2017), it was proposed to constrain the biases of the linear layer to be non-positive, i.e. $b_j \leqslant 0$. The main motivation was to guarantee that the origin is a root point of the function $f$. However, this is not the case, as the following simple counter-example will show. Suppose the bias $b = -\vec{1}$. Then the function $f(\vec{0}) = 0$, as $[W\vec{0} - \vec{1}]^+ = 0$, but the origin $\vec{0}$ is not a valid root point as the gradient is zero there. However, any input $\boldsymbol{x}$ with $f(\boldsymbol{x}) \geqslant 0$ will have a non-zero gradient and will therefore be in a different local region. In Figure 1, we visualized the local regions of a small 3-layered network for non-positive and unrestricted bias.

### 3.5 DTD for Deep Neural Neworks: The Training-Free Relevance Model

Applying the recursive Taylor decomposition to a one-layered network yielded a set of easily applied relevance propagation rules, which allowed to skip computing the root points explicitly. Of course, it would be desirable to skip the computation of roots for deep neural networks too. As solution, Montavon et al. (2017) proposed a so-called *training-free* relevance model. We follow the derivation from the review article (Montavon et al., 2018).

Let $R^{l+1}(\boldsymbol{a}_l)$ be the relevance computed for an upper layer. Montavon et al. (2017) then makes the following assumption:

**Assumption 1** (Positive Linear Relevance)**.** The relevance of the upper layer $R^{l+1}(\boldsymbol{a}_{l+1})$ can be written as $R^{l+1}(\boldsymbol{a}_{l+1}) = \boldsymbol{a}_{l+1} \odot \boldsymbol{c}_{l+1}$, where $\boldsymbol{c}_{l+1} \in \mathbb{R}^+$ should be a constant and positive vector.

As $R^{l+1}(\boldsymbol{a}_l) = \boldsymbol{c}_{l+1} \odot [W_l \boldsymbol{a}_l + b_l]^+$, we can construct a so-called relevance neuron:

$$\hat{R}^{l+1}(\boldsymbol{a}_{l+1}) = [\hat{W}_{l+1}\boldsymbol{a}_{l+1} + \hat{\boldsymbol{b}}_{l+1}]^+, \tag{14}$$

where we pulled $\boldsymbol{c}$ into the layer's parameters: $\hat{W}_{l+1} = W_{l+1} \odot C_{l+1}$ and where $C_{l+1} = [\boldsymbol{c}_{l+1}, \dots, \boldsymbol{c}_{l+1}]$ is a repeated version of $\boldsymbol{c}_{l+1}$ and $\hat{\boldsymbol{b}}_{l+1} = \boldsymbol{b}_{l+1} \odot \boldsymbol{c}_{l+1}$.

This formulation is similar to the relevance of the hidden layer of the one-layer network in equation 11. The difference is that the root point and search direction will be based on the modified weights $\hat{W}_{l+1}$ and $\hat{\boldsymbol{b}}_{l+1}$. Using $\hat{\boldsymbol{w}}_j = \hat{W}_{l+1_{[j:]}} = \boldsymbol{c}_{l+1_{[j]}} W_{l+1_{[j:]}}$, we can write the general relevance propagation rule of equation 13 as:

$$\hat{R}^l(\boldsymbol{a}_l) = \sum_{j=1}^{d} \left( \frac{\hat{\boldsymbol{w}}_j \odot \boldsymbol{v}_j}{\hat{\boldsymbol{w}}_j^T \boldsymbol{v}_j} \hat{R}_{[j]}^{l+1}(f_l(\boldsymbol{a}_l)) \right) = \sum_{j=1}^{d} \left( \frac{\boldsymbol{w}_j \odot \boldsymbol{v}_j}{\boldsymbol{w}_j^T \boldsymbol{v}_j} \hat{R}_{[j]}^{l+1}(f_l(\boldsymbol{a}_l)) \right), \tag{15}$$

where the $\boldsymbol{c}_{l+1_j}$ canceled out. The corresponding root point would be: $\tilde{\boldsymbol{a}}_l^{(j)} = \boldsymbol{a}_l - \frac{\boldsymbol{w}_j^T \boldsymbol{x} + b_j}{\boldsymbol{w}_j^T \boldsymbol{v}_j} \boldsymbol{v}_j$. Interestingly, this deviation recovers the one-layer case from equation 13. Thus, Montavon et al. (2018) argues that all

the rules from the linear case (Section 3.4) can be applied to a deep neural network too. This result can be easily extended to sum-pooling layers, as they are equivalent to a linear layer with the weights of value 1.

**Remark 3** (Is it correct that $\boldsymbol{c}_k$ is constant?)**.** A global constant $\boldsymbol{c}_k$ cannot exist, as changing the input vector can result a in totally different output, which would change the relevance magnitude. A local approximation of $\boldsymbol{c}_k$ could be correct if root points stays within the same local linear region where the function's gradient $\nabla f$ is locally constant.

**Remark 4** (**C5** no mechanism to enforce that the root point $\tilde{\boldsymbol{a}}_l^{(j)} \in N_{R^l}(\boldsymbol{a}_l)$? )**.** The corresponding root point to equation 15 would be: $\tilde{\boldsymbol{a}}_l^{(j)} = \boldsymbol{a}_l - \frac{\boldsymbol{w}_j^T \boldsymbol{x} + b_j}{\boldsymbol{w}_j^T \boldsymbol{v}_j} \boldsymbol{v}_j$. Will this root point be in the local region $N_f(\boldsymbol{a}_l)$ of $\boldsymbol{a}_l$? Probably not, as there is no mechanism enforcing this. We test this in more detail in the empirical evaluation.

### 3.5.1 Which Root Points To Choose For A Deep Neural Network?

In Section 3.4.1, we already discussed that there is no principled way to select the root points or corresponding rules. For deep neural networks, DTD Montavon et al. (2017) originally proposed to pick the $z^+$-rule for all layers except the first one. In a more recent work Montavon et al. (2019), it was argued to use a combination of LRP$_0$, LRP-$\varepsilon$ and the $\gamma$-rule. This was motivated by rather vague properties such as the "activations' entanglement", "spurious variations", and "spreading of relevance". It is concluded that: "*Overall, in order to apply LRP successfully on a new task, it is important to carefully inspect the properties of the neural network layers, and to ask the human what kind of explanation is most understandable for him.*" – Montavon et al. (2019, Sec. 10.3). Thus, the choice of the rules lies in the hands of the user who might choose any rule or root point.

Kohlbrenner et al. (2020) introduced a similar combination of rules as *LRP-Composite*. For the convolution layers, they used the $z^+$-rule (or the LRP$_{\alpha 1 \beta 0}$) and for the fully-connected layers the LRP$_0$. An improvement over using the $z^+$-rule for all layers, they found that this combination of rules did not suffer from class-insensitivity, i.e., the saliency map do change when the explained output class is changed. However, it must be noted that this combination relies on the particular properties of the convolutional neural network. Specifically, there is little information mixing between more distant locations. Furthermore, the explanations are still insensitive to the later convolutional layers: the $z^+$ rule creates a fixed saliency map for the convolutional layers, which, however, can be scaled by the output of the LRP$_0$-rule. For example, if the final convolutional output has shape (8, 8) than the saliency map can be scaled in an 8x8 grid.

## 4 Analysis of the Recursive Application of the Taylor Theorem

In the previous sections, we recapitulated how DTD applies the Taylor Theorem recursively to a one-layer and a deep neural network, and explained how the different propagation rules were derived. In this section, we provide a theoretical analysis of the recursive application of the Taylor Theorem. In particular, we study the definition 3 from Section 3.3. As this definition is the most general formulation of the DTD theory, we ensure that the results of our analysis are applicable to all the propagation rules and are also not caused by one specific approximation but are rather inherent to the recursive application of the Taylor theorem.

The following propositions are proven in the Appendix A. The main idea of the proof is to apply the product rule to equation 8 and then analyze the individual terms.

### 4.1 Size of admissable regions for the root points cannot be increased

**Proposition 1** (**C1**: Recursively applying the Taylor Theorem cannot increase the size of admissible regions)**.** *Given a ReLU network $f : \mathbb{R}^{d_1} \to \mathbb{R}_{\geq 0}^{d_{n+1}}$, recursive relevance functions $R^l(\boldsymbol{a}_l)$ with $l \in \{1, \ldots, n\}$ according to definition 3, and let $\xi$ index the explained logit, then it holds for the admissible region $N_{R^l}(\boldsymbol{a}_l)$ for the root points $\tilde{\boldsymbol{a}}_l^{(j)}$ of the relevance function $R^l$ that $N_{R^l}(\boldsymbol{a}_l) \subseteq N_{f_{n_\xi} \circ \ldots \circ f_l}(\boldsymbol{a}_l)$.*

As the valid region for root points is restricted by the network $f$, we then we cannot evade the local region. This motivates a simple empirical test in Section 5.1: for each root point, we can check whether it is contained

in the correct admissible region. This result questions the motivation that the distance $|\boldsymbol{x} - \tilde{\boldsymbol{x}}|$ might be small T from 3.2, as this distance remains bounded by the local linear region of the network.

## 4.2 Locally Constant Roots Imply Equivalence of Recursive Taylor and Gradient×Input

It is well known that $\text{LRP}_0$ is equivalent to gradient×input for ReLU networks. This was first noted in (Shrikumar et al., 2016) and later also in (Kindermans et al., 2016; Ancona et al., 2018). We proof the following generalization for the recursive application of the Taylor Theorem in Appendix A.1.

**Proposition 2 (C2).** *Let $f : \mathbb{R}^{d_1} \to \mathbb{R}^{d_{n+1}}_{\geqslant 0}$ be a ReLU network, $\xi$ be the index of the explained logit, and $R^l(\boldsymbol{a}_l)$ (with $l \in 1 \ldots n+1$) are recursive relevance functions according to definition 3. If the root points $\tilde{\boldsymbol{a}}_l(\boldsymbol{a}_l)$ are locally constant w.r.t. the layer's input ($\forall l \in 1 \ldots n : \partial \tilde{\boldsymbol{a}}_l / \partial \boldsymbol{a}_l = 0$), then:*

$$R(\boldsymbol{x}) = R(\tilde{\boldsymbol{x}}) + \nabla f_{[\xi]}(\boldsymbol{x}) \odot (\boldsymbol{x} - \tilde{\boldsymbol{x}}), \tag{16}$$

*where $\boldsymbol{x} = \boldsymbol{a}_1$ is the input vector and $R(\boldsymbol{x}) = R^1(\boldsymbol{x})$.*

The similarity with gradient×input can be seen when choosing a root point $\tilde{\boldsymbol{x}} = \boldsymbol{0}$ such that $R(\boldsymbol{0}) = \boldsymbol{0}$. Then, the resulting relevance would be $\nabla f_{[\xi]}(\boldsymbol{x}) \odot \boldsymbol{x}$.

A fixed root point for each linear region would be a valid and even desirable choice. For example, from an efficiency perspective, it would be preferable to search for a valid root point in each linear region only once. Or one might want to select the one root point corresponding to the lowest network output. We also want to emphasize that no continuous constraint for selecting the root points exists. Jumps at the boundaries between the linear region are allowed. This result contradicts DTD's motivation described in Section 3.2, as it explicitly aimed to find something more "stable" than the gradient.

## 4.3 Locally dependent root points

As a next case, we will look at the more general case of root points depending locally on the layer's input:

**Proposition 3 (C3).** *For a ReLU network $f : \mathbb{R}^{d_1} \to \mathbb{R}^{d_{n+1}}_{\geqslant 0}$ with $n$ layers, and layer activations $\boldsymbol{a}_l = f_{l-1}(\boldsymbol{a}_{l-1})$, the relevance functions $R^{l-1}(a_{l-1})$ of the recursive applications of the Taylor Theorem as given in equation 8 can be written as:*

$$R^{l-1}(a_{l-1}) = \sum_{j=1}^{d_l} \sum_{m=1}^{d_{l+1}} \left[ \left( \frac{\partial f_l(\boldsymbol{a}_l)}{\partial \boldsymbol{a}_{l_{[j]}}} - \frac{\partial \tilde{\boldsymbol{a}}_{l_{[j]}}^{(m)}(\boldsymbol{a}_l)}{\partial \boldsymbol{a}_l} \cdot \frac{\partial f_l(\boldsymbol{a}_l)}{\partial \boldsymbol{a}_l} \right) \cdot \frac{\partial R_{[m]}^{l+1}(f_l(\boldsymbol{a}_l))}{\partial f_l(\boldsymbol{a}_l)} \right] \odot \left( \boldsymbol{a}_{l-1} - \tilde{\boldsymbol{a}}_{l-1}^{(j)}(\boldsymbol{a}_{l-1}) \right), \tag{17}$$

The relevance function $R^{l-1}$ is determined by the next layer's Jacobian $\partial f_l(\boldsymbol{a}_l)/\partial \boldsymbol{a}_l$, and also a term including root point Jacobian $\partial \tilde{\boldsymbol{a}}_l^{(j)}/\partial \boldsymbol{a}_l$. Although some directions are recommended, the choice of root point is not restricted per se. It is merely recommended to choose it within the layer's input domain[2] and it should minimize the explained relevance. Any root point could be chosen, as long as it is from the linear region $N_{R_{[k]}^l}(\boldsymbol{a}_l)$. However, this also means that $R^{l-1}(\boldsymbol{a}_{l-1})$ can be influenced arbitrarily by the root point's Jacobian. Therefore, any explanation could be justified. A theory under which anything can be justified is clearly insufficient.

## 4.4 Why Not Use Analytic Activation Functions (Softplus)?

For ReLu networks, the Deep Taylor Decomposition suffers from the problem that the root point must be from the local linear region around the layer input $\boldsymbol{a}_l$. A possible solution would be to use an analytic activation function, e.g. the Softplus activation. This would allow to choose any root point in $\mathbb{R}^{d_l}$, although a sufficiently good approximation might require an unreasonable amount of higher-order terms. The main obstacle would be that with each decomposition, higher-order derivatives are accumulated:

---

[2] In Montavon et al. (2017, Section 4.1.), the different rules were selected based on the input domain. However, the $\gamma$ rule, introduced in a more recent work Montavon et al. (2019), can lead to root points outside the ReLU's input domain $\mathbb{R}^+$, e.g. for $\gamma = 0$ the root point is given by $\tilde{\boldsymbol{x}} = \boldsymbol{x} - \frac{\boldsymbol{x}}{\boldsymbol{w}^T \boldsymbol{x}} R^l(\boldsymbol{x})$ which can become negative for large relevance values $R^l(\boldsymbol{x})$.

Table 1: Empirical results of different DTD rules on a small neural network (3 layers, 10 input dimensions, 10 hidden dimensions). They show that the root points picked by the rules are not within the local region of the input, as each rule produced outputs below 100%. It is also the case that some root points will have the exact same network output as the original input.

| Evaluation \ Rule | $LRP_0$ | $\gamma = 1.0$ | $w^2$ | $z^+$ |
|---|---|---|---|---|
| Same local linear region [expected 100%] | 41.20% | 38.70% | 37.70% | 41.10% |
| Same network output [expected 0%] | 14.62% | 13.85% | 14.01% | 13.99% |

**Proposition 4 (C4).** *Let $f : \mathbb{R}^{d_1} \mapsto \mathbb{R}^{d_{n+1}}$ be a neural network, contains an analytic activation function, then each recursive application of the Taylor Theorem yields a higher-order derivative of the form:*

$$\frac{\partial}{\partial \boldsymbol{a}_l} \cdot \left[ \frac{\partial R_{[j]}^{l+1}(f_l(\boldsymbol{a}_l))}{\partial \boldsymbol{a}_l} \right]_{\boldsymbol{a}_l = \tilde{\boldsymbol{a}}_l^{(j)}(\boldsymbol{a}_l)} \cdot \left\{ \boldsymbol{a}_l - \tilde{\boldsymbol{a}}_l^{(j)}(\boldsymbol{a}_l) \right\}_{[k]}. \tag{18}$$

Thus, for a n-layered network, we would get n-ordered derivatives. The problem is that it is unclear how these chains of higher-order derivatives behave.

## 5 Experiments

### 5.1 (C6) DTD-Train-Free

We implemented the train-free DTD using an explicit computation of the root points. The network consists of 3 linear layers, each with a ReLU activation. The input and each layer has 10 dimensions. We initialized the network with random weights and used non-positive biases, as Montavon et al. (2017) suggested (even though we have shown that this has not the same consequences as claimed in Montavon et al. (2017). As we are only interested in disproving claims, it is acceptable to show that there exists one of neural network on which the DTD delivers inconsistent results. Therefore, we also did not train the network on any task.

We provide pseudocode for our implementation in Algorithm 2. The main simplifications of the implementation are that (1) the relevance of the higher layers is computed with the input of the layer and not the root point, and (2) the root points are computed using the search directions outlined in section 3.4. We tested our implementation against Captum's implementation of the DTD (Kokhlikyan et al., 2020) and found the deviation to be less than $1 \times 10^{-8}$.

Verifying that the two points are within the local region would require to show that the gradients are equal and that there is a path between the two points with all points on the path also having equal gradients. As the last part is more difficult to show, we only test the necessary condition of equal gradients. Therefore, we compare the gradient of the input with the gradient of the root points $|\nabla f(\boldsymbol{x}) - \nabla f(\tilde{\boldsymbol{x}})|$ on 1000 random inputs. The input points were sampled such that it has a network output greater than 0.1.

We reported the numerical results in Table 1. Less than 100% of all root points have gradient differences that are zero, thus root point exists which must be from a different local region. This violates Proposition 1, which requires all root points to be within the function's local region. Although we only show results on an exemplary 3-layered network, the situation would only be worse for more complex networks as the number of local regions increases exponentially with layer depth (Montufar et al., 2014).

As a second analysis, we tested how the root points influence the network output. One might assume that a root point will alter the network output. However, this is not always the case (see row "Same network output" in Table 1). At least, these root points do not explain the output of the neural network.

### 5.2 (C7) Applying Sanity Checks to (Arras et al., 2022)

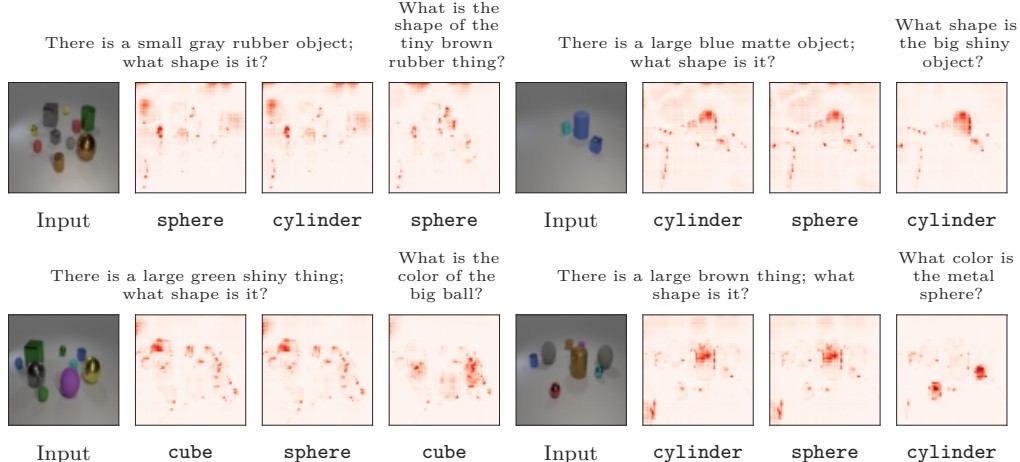

Figure 2: (1) Input images from the CLEVER-XAI dataset with the $LRP_{\alpha1\beta0}$ saliency maps computed for the (2) correct class, (3) an incorrect class, and (4) a different question. The original question is written above. The saliency maps do not change visually when a different output class is explained. However, changing the question highlights other regions.

A recent work (Arras et al., 2022) evaluated different saliency methods on the CLEVR VQA dataset using ground truth segmentation masks. Interestingly, they found $LRP_{\alpha1\beta0}$ (equivalent to the DTD $z^+$-rule) to highlight the object of interest particularly well: *"[...] a high connection between the relevance heatmaps and the target objects of each question"*. This finding seems to contradict Sixt et al. (2020), which found that $LRP_{\alpha1\beta0}$ becomes independent of the network's deeper layer. In Arras et al. (2022), it was therefore concluded: *"Maybe the phenomenon described in (Sixt et al., 2020) becomes predominant in the asymptotic case of a neural network with a very high number of layers, [...]"*.

A simple empirical test would have been to check if $LRP_{\alpha1\beta0}$'s saliency maps change when the network's last layer is changed. To perform this test, we replicated their setup and trained a relation network (Santoro et al., 2017) on the CLEVR V1.0 dataset Johnson et al. (2017). The network reached an accuracy of 93.47%, comparable to 93.3% (Arras et al., 2022) and 95.5% (Santoro et al., 2017). We included more details about the model in Appendix B.

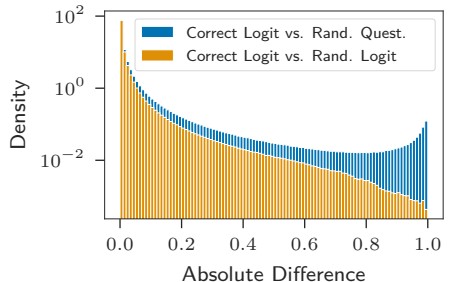

Figure 3: The histogram of absolute differences between the saliency maps for *Correct Logit vs. Random Question* and *Correct Logit vs. Random Logit.*

We then compared 1000 $LRP_{\alpha1\beta0}$'s saliency maps for the correct answer, an incorrect answer (but from the same category), and the correct answer but a different question. It is valid to ask for an explanation of a different class, for example, to understand which evidence is present for `sphere` instead of `cube`. The saliency maps were scaled to cover the range [0,1], and the differences were measured using the mean absolute difference. In Figure 3, a histogram of the differences is shown. While the saliency maps are very similar in both cases, there seems to be more variability in the question: for "correct logit vs. random question" there is an order of magnitude more pixels with a difference of $\approx 1$. When looking at the resulting saliency maps in Figure 2, one can see that the saliency maps differ quite significantly when changing the question. In contrast, the saliency map of the wrong answer does not change.

First, these results validate the claim in (Sixt et al., 2020) that $LRP_{\alpha1\beta0}$ is independent of the network's deeper layer. Second, they indicate that an information leakage between the question and $LRP_{\alpha1\beta0}$'s saliency maps is present.

The reason for this information leakage can be found in the specific architecture of the relation networks: pairs are formed between all feature-map locations of the convolutional output. As the feature map has shape

$(8, 8, 24)$, $64 \cdot 64$ pairs are formed (i.e., height$^2 \cdot$width$^2$). Additionally, the question embedding produced by an LSTM layer is concatenated to each pair. This yields triples of $(\boldsymbol{v}_{ij}, \boldsymbol{v}_{kl}, \boldsymbol{o}_{\text{LSTM}})$, where $i, j, k, l \in \{1, \ldots, 8\}$, and $\boldsymbol{v} \in \mathbb{R}^{8 \times 8 \times 24}$ is the convolutional stack's output. As the convolutional layers and the LSTM layer are trained together, their representations are aligned. Thus, changing the LSTM embedding will change the internal representation in the subsequent layers. For relevance locations, the question embedding will match better with the convolutional activation and, therefore will lead to a higher saliency map at relevant locations. However, the saliency maps will still become independent of the network's deeper layer.

The implications are substantial: for example, if the model's final layers' were fine-tuned on a new task, the LRP$_{\alpha 1 \beta 0}$ explanation would not change and could not be used to explain this model. Even worse, if your model was altered to predict spheres instead of cubes, the LRP explanation would not reflect this.

It is quite fascinating that the LRP$_{\alpha 1 \beta 0}$ explanations highlight the right object according to the ground truth, but fail to highlight evidence for the wrong object. This result also shows how difficult it is to evaluate explanation methods empirically.

## 6 Conclusion

We have shown that DTD, which has been cited as the theoretical foundation of numerous follow-up post-hoc explanation techniques (Ali et al., 2022; Binder et al., 2016; Kindermans et al., 2018; Arras et al., 2017; Hui & Binder, 2019; Huber et al., 2019; Eberle et al., 2020), exhibits serious flaws that explain why saliency maps created with these methods are independent of the output. From Sixt et al. (2020), we know that that the positive matrices produced by the $z^+$-rule will converge to a rank-1 matrix. These positive matrices stem from a specific selection of the root-point, and as the selection of the root-points is not restricted, the $z^+$-rule can be justified by the DTD theory, as every other explanation could be - by picking an appropriate root.

DTD as a theoretical framework for explanations is under-constraint, and can be considered insufficient. Caution must be used when using explanations derived from this theory. At the core of the problem, there is no restriction and little guidance on choosing the root points. Under certain conditions (constant root points), DTD reduces to backpropagating the gradient, albeit hidden behind a complex mathematical structure. In the other case (input-dependent root points), DTD leaves open a backdoor through which virtually any explanation can be created by crafting the root point's Jacobian accordingly. However, this again is obfuscated by the theory rather than made transparent.

Since its first ArXiv submission (Montavon et al., 2015), the DTD publication has been cited numerous times. Although even the authors have reported class-insensitive behavior (Kohlbrenner et al., 2020; Montavon et al., 2018)[3], follow-up works have readily used DTD's key concepts, motivated by the seemingly robust mathematical foundation, instead of searching for the underlying reasons. Furthermore, explanations based on DTD were used in various applications, for example, for validating their model (Andresen et al., 2020), gain insights into geoscientific questions (Toms et al., 2020), or conduct user studies (Alqaraawi et al., 2020).

While we were able to discover serious issues of DTD, we do not see a solution how to solve them. We therefore want to point out that other theoretically well-justified methods exist: *(Deletion of Information)* which information can be deleted without changing the network output? One approach uses noise (Schulz et al., 2020), other discrete deletion of the input (Macdonald et al., 2019). *(Testing prediction capabilities)*: We can test whether certain concepts are present in the network by trying to predict them from intermediate features (Kim et al., 2018). *(Model inversion)*: How would the input need to change to predict another class? This question can be answered using invertible models (Hvilshøj et al., 2021; Dombrowski et al., 2021) or conditional generative models (Singla et al., 2020). *(Simple Models)*: If a similar performance is achieved by a simpler, more interpretable model, why not simply use that? For example, (Zhang et al., 2021) replaced part of the network with a linear model. All these approaches do not come with a complicated mathematical superstructure, rather they have a simple and intuitive motivation.

---

[3] *"A reduced number of fully-connected layers avoids that the relevance, when redistributed backwards, looses its connection to the concept being predicted."* – Montavon et al. (2018, Sec 5.1.)

**Broader Impact Statement**

Although our work focuses on the theoretical foundations of a particular explanation method, we see broader implications of this work. Our work demonstrates that the theoretical foundation of explanation methods need rigorous analysis before they can support the trust that developers, users, and even regulatory bodies may put in it. This is especially important in the field of explainable AI since empirically evaluating explanations is difficult.

The field offers a variety of explanation methods, and ways to test the quality of explanations. We recommend using more than just one method and employing a range of metrics and user tests to make sure explanations are helpful in potentially critical use-cases such as medical decision making or the screening of job applications.

**Acknowledgement**

We want to thank the reviewers for their helpful feedback that improved this manuscript further. The computation were done on the Curta cluster provided by the Zedat, Freie Universtität Berlin (Bennett et al., 2020). Finally, we thank Jonas Köhler for discussions about the manuscript idea.

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

# A Proofs

## A.1 Proof of proposition 3 and 2

We will proof Propositions 2, 3, and 4 together.

We start with the partial derivative of the relevance function at layer $l$:

$$\frac{\partial R_{[k]}^l(\boldsymbol{a}_l)}{\partial \boldsymbol{a}_l} = \sum_{j=1}^{d_{l+1}} \frac{\partial}{\partial \boldsymbol{a}_l} \left( \left[ \frac{\partial R_{[j]}^{l+1}(f_l(\boldsymbol{a}_l))}{\partial \boldsymbol{a}_{l_{[k]}}} \right]_{\boldsymbol{a}_l = \tilde{\boldsymbol{a}}_l^{(j)}(\boldsymbol{a}_l)} \cdot \left\{ \boldsymbol{a}_l - \tilde{\boldsymbol{a}}_l^{(j)}(\boldsymbol{a}_l) \right\}_{[k]} \right) \tag{19}$$

$$= \sum_{j=1}^{d_{l+1}} \left( \frac{\partial \left( \boldsymbol{a}_{l_{[k]}} - \tilde{\boldsymbol{a}}_{l_{[k]}}^{(j)}(\boldsymbol{a}_l) \right)}{\partial \boldsymbol{a}_l} \cdot \left[ \frac{\partial R_{[j]}^{l+1}(f_l(\boldsymbol{a}_l))}{\partial \boldsymbol{a}_l} \right]_{\boldsymbol{a}_l = \tilde{\boldsymbol{a}}_l^{(j)}(\boldsymbol{a}_l)} \tag{20}$$

$$+ \underbrace{\frac{\partial}{\partial \boldsymbol{a}_l} \cdot \left[ \frac{\partial R_{[j]}^{l+1}(f_l(\boldsymbol{a}_l))}{\partial \boldsymbol{a}_l} \right]_{\boldsymbol{a}_l = \tilde{\boldsymbol{a}}_l^{(j)}(\boldsymbol{a}_l)} \cdot \left\{ \boldsymbol{a}_l - \tilde{\boldsymbol{a}}_l^{(j)}(\boldsymbol{a}_l) \right\}_{[k]}}_{= 0, \text{ for ReLU networks}} \right) \tag{21}$$

In this first step, we applied the product rule. For ReLU networks, the higher-order terms are zero. For other networks (Transformer, LSTMs), the higher-order terms will not be zero. The terms which are zero for ReLU networks are exactly the terms from Proposition 4.

In the next step, we will apply the chain rule and rewrite $\partial \boldsymbol{a}_{l_{[k]}}/\partial \boldsymbol{a}_l$ as the $k$-standard basis $e_k$ (a one-hot vector where the $k$-th dimension is 1):

$$\frac{\partial R_{[k]}^l(\boldsymbol{a}_l)}{\partial \boldsymbol{a}_l} = \sum_{j=1}^{d_{l+1}} \left( e_k - \frac{\partial \tilde{\boldsymbol{a}}_{l_{[k]}}^{(j)}(\boldsymbol{a}_l)}{\partial \boldsymbol{a}_l} \right) \cdot \left[ \frac{\partial f_l(\boldsymbol{a}_l)}{\partial \boldsymbol{a}_l} \cdot \frac{\partial R_{[j]}^{l+1}(f_l(\boldsymbol{a}_l))}{\partial f_l(\boldsymbol{a}_l)} \right]_{\boldsymbol{a}_l = \tilde{\boldsymbol{a}}_l^{(j)}(\boldsymbol{a}_l)}, \tag{22}$$

The next observation is that the gradients inside the $[\ldots]_{\boldsymbol{a}_l = \tilde{\boldsymbol{a}}_l^{(j)}(\boldsymbol{a}_l)}$ must be the same for $\boldsymbol{a}_l$ and the root point $\tilde{\boldsymbol{a}}_l^{(j)}$ as both are in the same local region of $f_l \circ R_{[j]}^{l+1}$. Therefore, we can safely drop the evaluation of the gradient at the root point ($[\ldots]_{\boldsymbol{a}_l = \tilde{\boldsymbol{a}}_l^{(j)}(\boldsymbol{a}_l)}$) and write:

$$\frac{\partial R_{[k]}^l(\boldsymbol{a}_l)}{\partial \boldsymbol{a}_l} = \sum_{j=1}^{d_{l+1}} \left( \frac{\partial f_l(\boldsymbol{a}_l)}{\partial \boldsymbol{a}_{l_{[k]}}} - \frac{\partial \tilde{\boldsymbol{a}}_{l_{[k]}}^{(j)}(\boldsymbol{a}_l)}{\partial \boldsymbol{a}_l} \cdot \frac{\partial f_l(\boldsymbol{a}_l)}{\partial \boldsymbol{a}_l} \right) \cdot \frac{\partial R_{[j]}^{l+1}(f_l(\boldsymbol{a}_l))}{\partial f_l(\boldsymbol{a}_l)} \tag{23}$$

Substituting this result into the definition of $R^{l-1}(\boldsymbol{a}_{l-1})$ from equation 8 yields the result of Proposition 3.

To proof proposition 2, we use $\partial \tilde{\boldsymbol{a}}_{l_{[k]}}^{(j)}(\boldsymbol{a}_l)/\partial \boldsymbol{a}_l = 0$ and get:

$$\frac{\partial R_{[k]}^l(\boldsymbol{a}_l)}{\partial \boldsymbol{a}_l} = \sum_{j=1}^{d_{l+1}} \frac{\partial f_l(\boldsymbol{a}_l)}{\partial \boldsymbol{a}_{l_{[k]}}} \cdot \frac{\partial R_{[j]}^{l+1}(f_l(\boldsymbol{a}_l))}{\partial f_l(\boldsymbol{a}_l)} \tag{24}$$

$$= \frac{\partial f_l(\boldsymbol{a}_l)}{\partial \boldsymbol{a}_{l_{[k]}}} \cdot \sum_{j=1}^{d_{l+1}} \frac{\partial R_{[j]}^{l+1}(f_l(\boldsymbol{a}_l))}{\partial f_l(\boldsymbol{a}_l)} \tag{25}$$

$$= \frac{\partial f_l(\boldsymbol{a}_l)}{\partial \boldsymbol{a}_{l_{[k]}}} \cdot \sum_{j=1}^{d_{l+1}} \frac{\partial f_{l+1}(\boldsymbol{a}_{l+1})}{\partial \boldsymbol{a}_{l+1_{[j]}}} \cdot \sum_{i=1}^{d_{l+2}} \frac{\partial R_{[i]}^{l+2}(f_{l+1}(\boldsymbol{a}_{l+1}))}{\partial f_{l+1}(\boldsymbol{a}_{l+1})} \tag{26}$$

$$= \frac{\partial f_l(\boldsymbol{a}_l)}{\partial \boldsymbol{a}_{l_{[k]}}} \cdot \frac{\partial f_{l+1}(\boldsymbol{a}_{l+1})}{\partial \boldsymbol{a}_{l+1}} \cdot \sum_{i=1}^{d_{l+2}} \frac{\partial R_{[i]}^{l+2}(f_{l+1}(\boldsymbol{a}_{l+1}))}{\partial f_{l+1}(\boldsymbol{a}_{l+1})} \tag{27}$$

$$= \frac{\partial f_l(\boldsymbol{a}_l)}{\partial \boldsymbol{a}_{l_{[k]}}} \cdot \frac{\partial f_{l+1}(\boldsymbol{a}_{l+1})}{\partial \boldsymbol{a}_{l+1}} \cdot \ldots \cdot \frac{\partial f_{n-1}^n(\boldsymbol{a}_{n-1})}{\partial \boldsymbol{a}_{n-1}} \cdot \sum_{i=1}^{d_{n+1}} \frac{\partial R_{[i]}^{n+1}\left(f_n^{n+1}(\boldsymbol{a}_n)\right)}{\partial f_n^{n+1}(\boldsymbol{a}_n)} \tag{28}$$

$$= \frac{\partial f_l(\boldsymbol{a}_l)}{\partial \boldsymbol{a}_{l_{[k]}}} \cdot \frac{\partial f_{l+1}(\boldsymbol{a}_{l+1})}{\partial \boldsymbol{a}_{l+1}} \cdot \ldots \cdot \frac{\partial f_{n-1}(\boldsymbol{a}_{n-1})}{\partial \boldsymbol{a}_{n-1}} \cdot \frac{\partial f_n(\boldsymbol{a}_n)}{\partial \boldsymbol{a}_n} \cdot e_\xi \tag{29}$$

$$= \nabla f_{l_{[\xi]}}(\boldsymbol{a}_l) \tag{30}$$

Substituting this into the relevance function of the input $R(\boldsymbol{x}) = R^1(\boldsymbol{a}_1)$ and using $\left.\frac{\partial R(\boldsymbol{x})}{\partial \boldsymbol{x}}\right|_{\boldsymbol{x}=\tilde{\boldsymbol{x}}(\boldsymbol{x})} = \frac{\partial R(\boldsymbol{x})}{\partial \boldsymbol{x}}$ (as $\tilde{\boldsymbol{x}}$ must be in the same linear region), yields:

$$R(\boldsymbol{x}) = R(\tilde{\boldsymbol{x}}(\boldsymbol{x})) + \left.\frac{\partial R(\boldsymbol{x})}{\partial \boldsymbol{x}}\right|_{\boldsymbol{x}=\tilde{\boldsymbol{x}}(\boldsymbol{x})} \odot (\boldsymbol{x} - \tilde{\boldsymbol{x}}) = R(\tilde{\boldsymbol{x}}(\boldsymbol{x})) + \nabla f_{[\xi]}(\boldsymbol{x}) \odot (\boldsymbol{x} - \tilde{\boldsymbol{x}}(\boldsymbol{x})), \tag{31}$$

which finished the proof of Proposition 2.

### A.2 Admissible Region for the root points of the Relevance Function

We now proof Proposition 1 which is restated here:

**Proposition 1** (C1: Recursive Taylor cannot increase the size of admissible regions) *Given a ReLU network* $f : \mathbb{R}^{d_1} \to \mathbb{R}^{d_{n+1}}_{\geq 0}$, *recursive relevance functions* $R^l(\boldsymbol{a}_l)$ *with* $l \in \{1, \ldots, n\}$ *according to definition 3, and let* $\xi$ *index the explained logit. Then it holds for the admissible region* $N_{R^l}(\boldsymbol{a}_l)$ *for the root points* $\tilde{\boldsymbol{a}}_l^{(j)}$ *of the relevance function* $R^l$ *that* $N_{R^l}(\boldsymbol{a}_l) \subseteq N_{f_{n_\xi} \circ \ldots \circ f_l}(\boldsymbol{a}_l)$.

Let $\tilde{\boldsymbol{a}}_1, \ldots, \tilde{\boldsymbol{a}}_n$ fix the root points. Proof by induction over the number of layers. We start with the induction base case at the final layer. There, we have $R^n(\tilde{\boldsymbol{a}}_n) = f_{[\xi]}(\tilde{\boldsymbol{a}}_n)$, which follows from the recursion base case. Clearly, $N_{R^n}(\tilde{\boldsymbol{a}}_n) \subseteq N_{f_{n_\xi}}(\tilde{\boldsymbol{a}}_n)$. Induction step: We assume $N_{R_{l+1}}(\tilde{\boldsymbol{a}}_{l+1}) \subseteq N_{f_{l+1}}(\tilde{\boldsymbol{a}}_{l+1})$. For the layer $l$, the root points must be valid for the function $R^{l+1}(f_l(\tilde{\boldsymbol{a}}_l))$. As we know that $N_{R_{l+1}}(\tilde{\boldsymbol{a}}_{l+1}) \subseteq N_{f_{l+1}}(\tilde{\boldsymbol{a}}_{l+1})$, it must also be the case that $N_{R^l}(\tilde{\boldsymbol{a}}_l) \subseteq N_{f_l}(\tilde{\boldsymbol{a}}_l)$.

## B  Details About the Relation Network for the CLEVR dataset

Our code-base builds upon a public available implementation of relation networks[4] and utilized Captum for computing $\text{LRP}_{\alpha 1 \beta 0}$ explanations Kokhlikyan et al. (2020). We also setup the CLEVR XAI dataset released on Github[5]

---

[4]https://github.com/rosinality/relation-networks-pytorch
[5]https://github.com/ahmedmagdiosman/clevr-xai/releases/tag/v1.0

## C    Pseudo-Code

### C.1    Full-backward DTD

---

**Algorithm 1** Pseudocode for the recursive application of the Taylor Theorem. The global state contains the following variables: $f_1, \ldots, f_n$ the layer functions of the network, $d_1, \ldots, d_{n+1}$ the dimension of the input to each layer, and $\xi$ the index of the output neuron.

---

$\quad$ **function** GET_RELEVANCE($l$: layer index, $\boldsymbol{a}_l$: the layer input)
$\quad\quad$ **if** $l = n + 1$ **then**
$\quad\quad\quad$ **return** $\boldsymbol{a}_{l_{[\xi]}}$
$\quad\quad$ **end if**
$\quad\quad$ $\tilde{\boldsymbol{a}}_l \leftarrow$ FIND_ROOT_POINT($f, l, \boldsymbol{a}_l$)
$\quad\quad$ $R^{l+1} \leftarrow$ GET_RELEVANCE($l + 1, f_l(\tilde{\boldsymbol{a}}_l)$)
$\quad\quad$ **for** $j \in 1 \ldots d_{l+1}$ **do**
$\quad\quad\quad$ $\tilde{\boldsymbol{a}}.\texttt{grad} \leftarrow 0$
$\quad\quad\quad$ $R^{l+1}_{[j]}.\texttt{backward}()$
$\quad\quad\quad$ $\boldsymbol{r}_j \leftarrow \tilde{\boldsymbol{a}}.\texttt{grad} \odot (\boldsymbol{a}_l - \tilde{\boldsymbol{a}}_l)$
$\quad\quad$ **end for**
$\quad\quad$ **return** $\sum_{j=1}^{d_{l+1}} \boldsymbol{r}_j$
$\quad$ **end function**

---

## C.2 Train-free DTD

---
**Algorithm 2** DTD Train-Free
---

**function** FIND_ROOT_POINT($l$: layer index, $\boldsymbol{a}_l$: the layer input, $R_{[j]}^{l+1}$: the relevance)
    $\boldsymbol{w}_j = W_{[j:]}$
    **if** $z^+$-rule **then**
        $\boldsymbol{v} = \boldsymbol{a}_l \odot \mathbb{1}_{\boldsymbol{w}_j \geqslant 0}$
    **else if** $w^2$-rule **then**
        $\boldsymbol{v} = \boldsymbol{w}_j$
    **else if** $\gamma$-rule **then**
        $\boldsymbol{v} = \boldsymbol{a}_l(1 + \gamma \mathbb{1}_{\boldsymbol{w}_j \geqslant 0})$
        $\ldots$
    **end if**
    $t = R_{[j]}^{l+1}/(\boldsymbol{w}_j^T \boldsymbol{v})$
    **return** $\boldsymbol{a}_l - t\boldsymbol{v}$                 $\triangleright$ Ensures that $\boldsymbol{w}_j^T(\boldsymbol{a}_l - \tilde{\boldsymbol{a}}_l) = \boldsymbol{w}_j^T \frac{R_{[j]}}{\boldsymbol{w}_j^T \boldsymbol{v}}\boldsymbol{v} = R_{[j]}$
**end function**

**function** GET_RELEVANCE($l$: layer index, $\boldsymbol{a}_l$: the layer input)
    **if** $l = n + 1$ **then**
        **return** $\boldsymbol{a}_{l_{[\xi]}}$
    **end if**
    $R^{l+1} \leftarrow$ GET_RELEVANCE($l + 1, f_l(\boldsymbol{a}_l)$)           $\triangleright$ relevance of input instead of root point $\tilde{\boldsymbol{a}}_l$
    **for** $j \in 1 \ldots d_{l+1}$ **do**
        $\tilde{\boldsymbol{a}}_l^{(j)} \leftarrow$ FIND_ROOT_POINT($f, l, \boldsymbol{a}_l, R_{[j]}^{l+1}$)
        $\tilde{\boldsymbol{a}}.\texttt{grad} \leftarrow 0$
        $\boldsymbol{o} \leftarrow [W_{l_{[j]}}\boldsymbol{a}_l^{(j)} + \boldsymbol{b}_{l_{[j]}}]$
        $\boldsymbol{o}.\texttt{backward}()$
        $\boldsymbol{r}_j \leftarrow \tilde{\boldsymbol{a}}.\texttt{grad} \odot (\boldsymbol{a}_l - \tilde{\boldsymbol{a}}_l)$
    **end for**
    **return** $\sum_{j=1}^{d_{l+1}} \boldsymbol{r}_j$
**end function**

---

