# OpenReview forum: "A Rigorous Study Of The Deep Taylor Decomposition"
_TMLR — Accepted by TMLR_

### Review · Reviewer_nb1x · 2022-09-15

**Summary Of Contributions:**

In the paper "A Rigorous Study of the Deep Taylor Decomposition", the authors provide some technical work related to the generation of post-hoc explanations from trained machine learning models.  The authors point out that some existing methods create explanations remain constant even if the last layer is randomized.  The authors note that the Deep Taylor Decomposition is used to provide theoretical underpinning of methods like this and establish technical claims relating root points associated with the Deep Taylor Decomposition to the explanations produced.


**Broader Impact Concerns:**

None.

**Requested Changes:**

I recommend a substantial rewrite of the introduction (and possibly abstract) so that a reader who is not thoroughly familiar with this subfield can understand the context and the conclusions drawn.

I also recommend rewriting throughout the whole paper.  It is hard to distinguish what is background material versus what is new.  It is also hard to figure out what the authors are intending to do in each section.  It would help if the authors at each section led with a paragraph explaining what was going to happen in that section.

Without substantial rewriting I do not feel I will understand the work well enough to be able to recommend for acceptance.

**Strengths And Weaknesses:**

A strength of this paper is that it attempts to provide a more thorough theoretical understanding of explanation methods, which are an important problem in ML.

A weakness of this paper is that the writing was not clear to me as a non expert in this subfield.  It may be the case that the writing is appropriate for readers who are experienced in this subfield, but I am not such a reader.  Here are some aspects that were unclear to me:
- What is the nature of explanations provided by the studied methods?
- What is the z+ rule and what does it mean to say it will "remain the same even if the last layer is randomized"?
- What is the logic use in relating "flaws" in the DTD theory to theoretical explanations?  Is the contribution of this paper that certain scenarios do not satisfy assumptions necessary in existing theoretical work for DTD-theory to apply?
- In (T1) the authors say they show that "the root points must be contained in the same linear region as the input".   What do they mean by saying "must be"?
- The paper mentions "root points", but it is not clear to me what quantity has roots that are being studied.

These weaknesses make it difficult for me to assess whether the conclusions support the claims.

---

> ### Author Response · Authors · 2022-09-30
> **Rebuttal**
>
> We thank the reviewer for their feedback and understand that our paper is challenging to access for someone outside the field. As discussed in the general answer, we rewrote the introduction and restructured the paper organization. We also added a footnote to define the *root point*. We hope that these changes make our work more accessible. But, of course, our paper remains technical and requires a certain knowledge about the field and mathematics, which we cannot simplify away.

---

### Review · Reviewer_rZ5T · 2022-09-17

**Summary Of Contributions:**

This paper undertakes the study of the method called Deep Taylor Expansion DTE), which is a strategy to provide some kind of post hoc explanations for the output of neural networks. In particular, the authors present the main ideas (expanding the output in terms of a Taylor expansion), explain how this is simplified when the root points are in the same linear regions, and discuss different strategies on how to do this. Contrary to what one would expect, these different choices are shown to be problematic (e.g. resulting in relevance vectors that can be independent of input, or that can be obtained to justified any output). The empirical demonstration furthermore shows that in practice some of the underlying assumptions -such as the root corresponding to the same linear region- don't really hold either.

In a nutshell, this work attempts to shed some light onto the DTE, clarifying some concepts while disproving some common "myths" about this approach.

**Broader Impact Concerns:**

While not strictly necessary, I would recommend the authors include a Broader Impact Concerns section. This work does not represent any concerns per se, but I think this would be a good opportunity for the authors to raise concerns on using explanation methods that lack the necessary careful and precise theoretical guarantees.

**Requested Changes:**

I very much appreciate this contribution, and the authors' intention to clarify what certain popular approaches do, and what they do not do. I only have a few minor comments:

- As I read through the paper, I thought that some comments on other theoretically-motivated methods for explainability would be contribute to the broader picture of the field. Later, almost at the end, I discovered that the authors do in fact have a section on Related Works which addresses these points. I believe it might be better to relocate that section to near the beginning, perhaps after the introduction. In that way, the reader will understand that several other principled methods exist, and that this paper will only study one family of them (DTE). Explaining this explicitly might provide a nice context for the reader to better understand the contribution of this paper.

- I think there are some works that the authors might find relevant for this paper. E.g.: [Do Input Gradients Highlight Discriminative Features?, Shah et al, 2021] provides very interesting insights showing that methods based on gradients can be provably incorrect - which supports the finding of this paper. Furthermore, the work [From Shapley back to Pearson: Hypothesis Testing via the Shapley Value, Teneggi et al, 2022] shows that Shapley values, contrary to previous interpretations and in a similar spirit to this paper, provides a formal notion of statistical importance by mathematically analyzing what those coefficients estimate.

- In Section 2.2, the authors refer to the exponential number of linear regions of deep nets. While this is true in the worst case, note that more recent analysis has shown that these are significantly lesser, and can even become linear [Complexity of Linear Regions in Deep Networks, Hanin & Rolnick, 2019].

- After Eq (11), in the expression $R^h$, $h$ should be bold.
- First line in section 3.2.1, "principle" -> "principled".
- Last line in first paragraph in Sec 3.2.1,  "choice" -> "choose"
- Second paragraph in Sec 3.2.1, could the authors clarify what they mean exactly by "class insensitivity"?
- Proposition 2: The first sentence is missing a subject. Maybe the authors meant "Let f be a Relu network...", or else the sentence should not break before continuing with the assumptions on the root points.
- First line after Prop 2, the zero in $R(0)$ should be bold.
- Last paragraph on page 8, the authors write that "It is merely recommended to choose it within the layer’s input domain". What do the authors mean exactly? Anything on $\mathbb{R}^{d_l}$? Or perhaps the authors meant "within the layer's input linear region"?
- Sec 5.1, the authors note that if the difference $||\nabla f(x) - \nabla f(\tilde{x}) ||=0$, the root points are assumed to be in the same linear region. Rather than "assuming" this, the authors might want to explain that this condition is necessary but not sufficient. Thus, showing that this is not the case, suffices for the argument they are trying to make.
- Close to the end of page 9: "Less than 100% of all root points have gradient differences are zero,.." -> "Less than 100% of all root points have gradient differences *that* are zero,..".


**Strengths And Weaknesses:**

Strneghts:
- The paper is clearly written, including an overview of the overall framework and ideas
- The paper is well motivated, and has an important purpose: if explanations methods are to provide further understanding about the predictions of NNs, it is fundamental that this is done in a rigorous and careful way, and not resulting to loosely motivated 'hacks'.
- The contribution is simple and important.

Weaknesses:
- Very small reorganization of content, and rephrasing of some parts might be good (see below).
- Some related references missing (see below).

---

> ### Author Response · Authors · 2022-09-30
> **Rebuttal**
>
> We thank the reviewer for their feedback and the overall positive assessment. As stated in the general answer, we hope that the changes to the introduction and organization addressed your criticism.
>
> We thank you for pointing us to the interesting works, which we now mention in the related work section. We also appreciated the list of minor issues in the manuscript such as missing boldness or grammar. We fixed them all. We also added a broader impact section based on your suggestion.

---

> > ### Comment · Reviewer_rZ5T · 2022-10-04
> > **Thank you for the response**
> >
> > I appreciate the authors' responses to my comments. I think the paper is in a better condition to be published at TMLR.

---

### Review · Reviewer_u2ZG · 2022-09-19

**Summary Of Contributions:**

This paper provides a very technical theoretical study on the shortcomings of the Deep Taylor Decomposition (DTD) as a post-hoc explanation method for deep learning. Specifically, this work builds on prior studies that showed that the DTD method could produce arbitrarily different explanations depending on the root point specified by the user. The main contribution of this work is therefore a series of theoretical arguments that scrutinize this behavior and give a technical explanation for why it happens. In short, the authors show analytically that the DTD method can be arbitrarily manipulated to produce any explanation through the effect of the intermediatee layer Jacobians with respect to the root points, and empirically, that the common heuristics used in practice to identify valid root points for the DTD method do not actually satisfy its main assumptions. This series of arguments leads the authors to conclude that the DTD method is not a theoretically sound post-hoc explanation method as it is (i) underspecified in its construction, and (ii) its main assumptions are violated by the heuristics used in practice.

**Broader Impact Concerns:**

I see no clear broader impact concerns derived from this very theoretical and technical work.

**Requested Changes:**

Overall, I think this paper meets the requirements to be published at TMLR: (i) it is technically sound, and (ii) it may appeal to some people in the community (although in this particular case, it might be a narrow set of the community interested in the technical details of the DTD method).

In any case, I still bellieve the paper could be easily improved, mostly by improving its introduction, structure and notation, as I listed under "weaknesses" before. Running the experiments on a less toy setting might also make the paper appeal to a broader audience (although I personally believe this is not strictly necessary).

Another way the authors could strengthen the appeal of their technical derivations is if they used their derived expressions explicitly to break a real trained network's DTD explanation by identifying root points that make the computed relevance independent of the output. This would be similar to the methods that compute adversarial examples on a model's explanations but in this case the authors would make use of their analytically derived expressions, thus providing further insights about the structure of these failure modes.

**Strengths And Weaknesses:**

# Strengths

1. **A solid technical argumentation based on simple math**: The main strength of this work is that it provides theoretically solid arguments to settle a previously observed empirical phenomenon. Although very technical and specific to the DTD technique, the arguments are easy to follow and do not require heavy mathematical machinery. The fact that the root points influence the explanations through the Jacobian of the layer's output might look obvious in retrospect, but it is nice to see some work that provides analytical expressions for this dependence.

2. **Concrete message**: I appreciate that this work focuses on a very specific problem with a clear objective: "Analytically explaining how the DTD method is underspecified and why can be made to fail in practice"

# Weaknesses

1. **Introduction is too technical**: Although I appreciate that this work is focused on a technical topic and goes straight to its point, I find the starting section to be too technical and hard to follow on a first read due to the use of very specific DTD jargon which has not been introduced before right from the start (e.g., $z^+$- rule or LRP).

2. **Abuse of heavy notation**: I find that some sections of the paper could be better off with some polishing of the notation to reduce the overuse of incoherent subscripts and upperscripts. For example, in Section 2 the relevances of each layer are denoted as $R^l$ while later on  Section 3 they are denoted as $R^f$, $R^{x}$ and $R^{h}$.  The distinct use of "Tn" or "An" to denote the different arguments is also a bit confusing.

3. **Hard-to-follow storyline**: The structure of the paper is sometimes a bit hard to follow due to the constant use of technical detours to introduce new concepts. In this regard, I believe that having a less technical introduction where the main issues are broadly discussed followed by a better structured discussion of the paper organization could help with this.

4. **(Minor) Unnecesarily naive experiments**: Although I do not think it is important for the paper's argumentation, I believe there is no reason for the experiments of this work to be performed on such a toy setting. The authors could very easily provide at almost no computational cost or engineering effort the same experiment using some standard classification dataset such as CIFAR-10 and a popular architecture like ResNet18.

---

> ### Author Response · Authors · 2022-09-30
> **Rebuttal**
>
> First, we want to thank the reviewer u2ZG for the feedback and their time.  Regarding the story line and introduction, please see our general remarks.
>
> > **Abuse of heavy notation***: I find that some sections of the paper could be better off with some polishing of the notation to reduce the overuse of incoherent subscripts and upperscripts. For example, in Section 2 the relevances of each layer are denoted as $R^f$ while later on Section 3 they are denoted as $R^f$ , $R^x$ and $R^h$. The distinct use of "Tn" or "An" to denote the different arguments is also a bit confusing.*
>
> We agree that in the section “Deep Taylor Decomposition For A One-Layer Network” the usage of $R^f$ , $R^x$ and $R^h$ is a slight abuse of notation. However, we believe it is easy to understand that $R^x$ corresponds to the relevance computed for the variable $x$ and analogously for $f$ and $h$. The alternative to use $R^1$, $R^2$, $R^3$ would be rather unintuitive. We added a comment about the differences in notation between the one-layer and the multi-layered case.
>
> We do understand that some equations have many super- and subscripts. It is hard to avoid them without sacrificing explicitness.
> For example, in the Definition 3: Recursive Taylor, we need to index the layer ($l$), the explained dimension of the upper layer $j$.
> This leads to expressions such as $ R^{l+1}_{[j]} $  dimension $ j $ of the relevance of layer $ l+1 $
> and $\tilde a_l^{(j)}$ root-point for layer $l$ and the $j$-th dimension.
>
> The usage of superscript for $R^l$ is because we often index the dimensions of $R^l$ as in $R^l_{[j]}$. The alternative to use superscript also for all variables would also not make it particularly nicer.
>
>
> > **(Minor) Unnecesarily naive experiments**: Although I do not think it is important for the paper's argumentation, I believe there is no reason for the experiments of this work to be performed on such a toy setting. The authors could very easily provide at almost no computational cost or engineering effort the same experiment using some standard classification dataset such as CIFAR-10 and a popular architecture like ResNet18.
>
> As you already indicated, we would not expect any additional insights, as it is clear that if it fails in this toy setting, it would be only worse for deep networks. Furthermore, our implementation is not so easily extendable to convolutional, batch norm, and shortcut layers. The main reason is that we construct the root-points explicitly, contrary to practical implementations which apply the rules directly. While it would be possible to extend our implementation by expressing these operations as matrix products, it would require a substantial effort.
>
>
>
> > Another way the authors could strengthen the appeal of their technical derivations is if they used their derived expressions explicitly to break a real trained network's DTD explanation by identifying root points that make the computed relevance independent of the output. This would be similar to the methods that compute adversarial examples on a model's explanations but in this case the authors would make use of their analytically derived expressions, thus providing further insights about the structure of these failure modes.
>
> It is correct that one could use our theoretical insights to craft custom explanations. We also did a first experiment in this direction to set all internal activations to zero by an appropriate choice of the root-points, but finally decided against including such an experiment. We would need to conduct such an experiment on a convolutional network to provide interesting results, which would require a major extension of our current code base as discussed above.
> We see our manuscript as a theoretical contribution to create the foundation for future work to investigate such practical manipulations.

---

> > ### Comment · Reviewer_u2ZG · 2022-10-04
> > **Stronger paper after rebuttal**
> >
> > Thank you very much for your response and for having taken the time to rewrite the paper based on the suggestions of all the reviewers. Personally, I think the new manuscript, with a much more accessible introduction, is certainly much stronger and deserves to be accepted. The new section 3 is now maybe a bit too long for my taste, and I feel it could be split into two shorter sections: Preliminaries (up to 3.3) and DTD formulation (including 3.4 and 3.5); but this is really just a personal opinion.
> >
> > Regarding the fact that the experiments are provided in a toy setting, I completely follow the author arguments, and as I said in my original review, I do not think it is necessary to go beyond the toy setting. I was just trying to give a suggestion for something which (although it would require a bit of work) could certainly boost the impact of the paper. This is again, just an opinion, and does not affect my main assessment of the paper.
> >
> > Overall, I believe this paper deserves to be accepted to TMLR and I will recommend its acceptance.

---

### Author Response · Authors · 2022-09-30
**General Answer to the Reviews**

First, we want to thank the reviewers for their time, the overall positive feedback, and the helpful criticism. They found our manuscript "solid technical argumentation based on simple math", "clearly written", and "well motivated". All reviewers agreed that it is a significant contribution, and the novelty of our work was also not disputed.

The main criticism shared by all reviewers was the technical introduction and the unclear structure of the paper. We addressed that by rewriting the introduction, including a broader motivation. Furthermore, we reorganized the paper by moving the related work section after the introduction and combining sections "Applying Taylor Theorem Recursively" and "Deep Taylor Decomposition" into a single "Background" section.

We also expanded the transitions between the individual subsections to give the reader more context.

---

### Decision · Action_Editors · 2022-11-03

**Recommendation:** Accept as is

**Comment:**

This paper undertakes the study of the method called Deep Taylor Expansion DTE), which is a strategy to provide some kind of post hoc explanations for the output of neural networks.   During the initial review period, reviewers raised some concerns about this paper which later have been addressed by the author's rebuttal.   All reviewers believe this paper meets the requirements to be published at TMLR.  Reviewer rZ5T believes that the related works section could be broader, and the authors could have done a better job at situating their contribution in light of other, theoretically-sound, explanation methods.  I hope authors could make improvements in their camera-ready version.




**Audience:**

Yes.

**Claims And Evidence:**

Yes.